# Effect of Preharvest 1-MCP Treatment on the Flesh Firmness of 'Rojo Brillante' Persimmon

Nariane Q. Vilhena [1] , Rebeca Gil [1], Mario Vendrell [2] and Alejandra Salvador [1],*

1 Postharvest Department, Instituto Valenciano de Investigaciones Agrarias, 46113 Valencia, Spain; quaresma_nar@gva.es (N.Q.V.); gil_reb@gva.es (R.G.)
2 Cooperativa Agrícola Nuestra Señora del Oreto, 46250 Valencia, Spain; mvendrell@cansocoopv.es
* Correspondence: salvador_ale@gva.es

**Abstract:** This study investigated the effect of preharvest 1-MCP treatment on maintaining 'Rojo Brillante' persimmon firmness. Early in the season, preharvest 1-MCP was applied 1, 7 and 10 days after ethephon treatment. The fruit firmness was evaluated during three different harvests and after the commercialization period of 3 d at 3 °C, plus 6 d at 20 °C. Late in the season, 1-MCP was applied 3 days before harvest in the fruit treated with gibberellic acid (GA) and then cold-stored for up to 60 days, plus a 6-day shelf life at 20 °C. The results showed that preharvest 1-MCP delayed the fruit softening induced by ethephon during the harvest period, and was the most effective treatment when performed 1 day after ethephon application. Therefore, preharvest 1-MCP extended the harvest period of ethephon-treated fruit. At the end of the season, preharvest 1-MCP had the same effect on maintaining the fruit firmness as the commercial postharvest application.

**Keywords:** *Diospyros kaki* Thunb.; 1-methylcyclopropene; ethylene; fruit firmness





## 1. Introduction

'Rojo Brillante' persimmon (*Diospyros kaki* Thunb.) is the main cultivar produced in the Mediterranean region, and is commercialized as fruit with high firmness values after being subjected to deastringency treatment at high $CO_2$ concentrations [1]. Therefore, flesh firmness is the main attribute that must be maintained during the postharvest period [2].

As the persimmon maturation period is short, fruit cold storage is necessary to allow for exportation and to destine part of production to cover commercial demands at the end of the season. However, persimmon is sensitive to low temperature and develops chilling injury (CI) symptoms, such as flesh gelling and firmness loss [3]. Previous studies have reported how the postharvest application of 1-methylcyclopropene (1-MCP), an innocuous gas used at very low concentrations, inhibits ethylene action by binding to ethylene receptors, which alleviates CI symptoms in most persimmon varieties [4,5]. Therefore, 1-MCP is routinely applied in industry for cold-stored persimmon fruit.

To date, 1-MCP is applied to persimmon as a postharvest treatment prior to cold storage. However, preharvest 1-MCP (Harvista®, Philadelphia, PA, USA) treatment, applied as a liquid spray to trees, is a reported novel option for maintaining fruit quality throughout the postharvest and replaces postharvest treatment in some crops, such as apples or pears [6–9]. In persimmon, information about the effect of preharvest 1-MCP application is scarce. Only one study about the 'Fuyu' cultivar has reported a positive effect of this treatment on retarding fruit maturity on trees [10].

In the specific case of 'Rojo Brillante', it would be interesting to know the effect of applying preharvest 1-MCP in different scenarios; on the one hand, at the end of the season to the fruit destined for cold storage. In this case, fruit are usually treated with gibberellic acid (GA) during the preharvest to delay fruit ripening [11], whereas postharvest 1-MCP is applied prior to cold storage. On the other hand, the effect of preharvest 1-MCP should be tested early in the season, in fruit treated with ethephon to advance maturity. When

ethephon is applied, the fruit harvesting period is short because they begin to ripen with a consequent firmness loss. Moreover, ethephon-treated fruit must be marketed quickly after harvest. Thus applying 1-MCP is often necessary to maintain commercial firmness values during the marketing period.

In this context, the present study investigated the effect of preharvest 1-MCP application on maintaining the firmness of 'Rojo Brillante' persimmon in these two scenarios: (1) fruit treated with ethephon early in the season; (2) fruit treated with GA late in the season.

## 2. Materials and Methods

### 2.1. Fruit Material

2.1.1. Experiment 1: Applying Preharvest 1-MCP to Ethephon-Treated Fruit

This study was conducted in two commercial 'Rojo Brillante' persimmon orchards located in Alcudia (Valencia, Spain) (lat. 39°11′25.5″ N, long. 0°32′43.0″ W and lat. 39°12′25.5″ N, long. 0°28′28.2″ W). Average temperature and relative humidity during the experiment period were taken from the IVIA weather station and ranged from 13.5 °C to 22.01 °C and 43.1% to 94.8%, respectively.

In each orchard, four rows of six trees were randomly taken for subsequent treatments. All of the trees in both orchards were ethephon-treated (0.08 cm$^3$ L$^{-1}$) (Fruitel®, Bayer Cropscience S.L., Leverkusen, Germany) under commercial conditions on October 5, when the fruit color index was −1.75 (CI = 1000 a/Lb, 'L', 'a', 'b' Hunter parameters). The trees of three rows were sprayed with preharvest 1-MCP (pre-MCP, 12 g L$^{-1}$) (Harvista®, Agrofresh Inc., Philadelphia, PA, USA) on days 1 (pre-MCP-1d), 7 (pre-MCP-7d) and 10 (pre-MCP-10d) after ethephon application, respectively. The fourth row was not treated with preharvest 1-MCP (CTL).

Three harvests took place: the first one the day after the last pre-MCP application (16 October), and the following harvests on 30 October and 10 November. On each harvest date, 150 fruit per treatment were picked. One lot of 50 fruit was characterized at harvest. The other two lots of 50 fruit were submitted to a simulated 3-day commercialization period at 3 °C, plus 6 days at 20 °C, with or without the postharvest 1-MCP treatment (post-MCP). Postharvest 1-MCP (Smartfresh®, Agrofresh Inc.) was applied under commercial conditions (0.5 µL L$^{-1}$ for 24 h) in cold chambers [12].

2.1.2. Experiment 2: Applying Preharvest 1-MCP to Gibberellic Acid-Treated Fruit

This study was performed in the other two commercial 'Rojo Brillante' orchards in Alcudia (Valencia, Spain) (lat. 39°10′51.2″ N, long. 0°30′05.3″ W and lat. 39°10′51.1″ N, long. 0°30′05.9″ W). Average temperature and relative humidity during the experiment period were taken from the IVIA weather station and ranged from 10.1 °C to 23 °C and 35% to 94.8%, respectively.

In each orchard, four rows of six trees were randomly taken for subsequent treatments. The trees of two rows were sprayed with GA (30 µL L$^{-1}$) (Berelex® 40 SG, Kenogard S.A., Barcelona, Spain) on September 25, when the fruit skin color index came close to −6 (one GA treatment (GA1)). The trees of the other two rows were sprayed with two GA treatments (GA2) on September 25 and October 15. Three days before harvesting, one row of GA1 and one row of GA2 were sprayed with pre-MCP (22 g L$^{-1}$). In accordance with commercial criteria, the fruit from GA1 and GA2 were harvested on November 16 and 23 November, respectively.

After harvesting, lots of 50 fruits were formed. One lot per treatment was evaluated at harvest. In order to compare the effect of 1-MCP applied at pre- or postharvest, part of the lots was treated with post-MCP (0.5 µL L$^{-1}$ for 24 h) prior to cold storage. This gave six different treatments:

(1)    GA1 (fruit treated once with GA);
(2)    GA1 + pre-MCP (fruit treated once with GA + preharvest 1-MCP);
(3)    GA1 + post-MCP (fruit treated once with GA + postharvest 1-MCP);

(4)   GA2 (fruit treated twice with GA);
(5)   GA2 + pre-MCP (fruit treated twice with GA + preharvest 1-MCP);
(6)   GA2 + post-MCP (fruit treated twice with GA + postharvest 1-MCP).

One lot of each treatment was evaluated after 20, 40 or 60 days at 0 °C, plus 6 days at 20 °C, to simulate the shelf-life period.

### 2.2. Determinations

At harvest and after the different storage periods, flesh firmness was determined by a texturometer (Instron Corp., mod. 4301, Canton, MA, USA) using an 8 mm diameter punch. The results were expressed as the force (N) needed to break the pulp in the equatorial zone, from which, skin had been previously removed.

Data were subjected to analyses of variance (ANOVA). The multiple comparisons between means were determined by the LSD test ($p \leq 0.05$) with the Statgraphics Centurion XVII.I software application (Manugistics, Inc., Rockville, MD, USA).

## 3. Results

### 3.1. Effect of Applying Preharvest 1-MCP on Ethephon-Treated Persimmon

On the three harvest dates, an effect of pre-MCP treatments was found on the flesh firmness. No differences between the orchards were observed. The lowest firmness values were for the control fruit (CTL) on all of the harvest dates, and no large differences appeared among the three pre-MCP treatments (Figure 1). Only in the second harvest (30 October) did the pre-MCP-10d fruit have slightly lower values than the pre-MCP-1d and pre-MCP-7d fruit. On the third harvest date, the CTL fruit obtained values of 23.2 N, and the pre-MCP-treated fruit still had values close to 35 N regardless of the date when the treatment was applied.

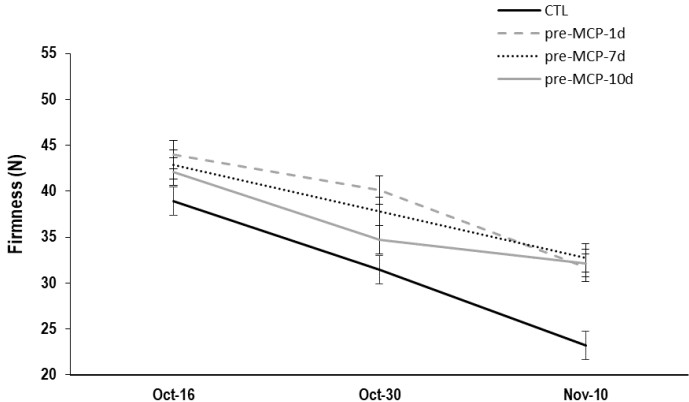

**Figure 1.** Effect of applying preharvest 1-MCP on days 1 (pre-MCP-1d), 7 (pre-MCP-7d) or 10 (pre-MCP-10d) after ethephon treatment on 'Rojo Brillante' persimmon flesh firmness on three harvest dates. CTL is the fruit not treated with preharvest 1-MCP. Vertical bars represent the least significant differences (LSD) intervals ($p \leq 0.05$).

After the commercialization period, in the fruit harvested on 16 October, the pre-MCP-1d and pre-MCP-7d fruit without post-MCP had the highest values, which were close to 40 N (Table 1). The pre-MCP-10d fruit obtained lower firmness values of 36 N, which were higher than those of the CTL treatment (29.29 N). A similar effect was observed for the following harvests: while the CTL fruit presented firmness values close to 13 N, the pre-MCP-treated fruit had values above 20 N in all cases.

**Table 1.** Effect of preharvest 1-MCP applied on days 1 (pre-MCP-1d), 7 (pre-MCP-7d) or 10 (pre-MCP-10d) after ethephon treatment and postharvest 1-MCP (post-MCP) on 'Rojo Brillante' persimmon flesh firmness after the commercialization period (3d at 3 °C plus 6 d at 20 °C) on three harvest dates. CTL is the fruit not treated with pre-MCP.

| | Harvest Date | | | | | |
| | 16 October | | 30 October | | 10 November | |
| | No Post-MCP | With Post-MCP | No Post-MCP | With Post-MCP | No Post-MCP | With Post-MCP |
|---|---|---|---|---|---|---|
| CTL | 29.29 [aA] | 36.18 [aB] | 13.40 [aA] | 24.88 [aB] | 13.0 [aA] | 18.46 [aA] |
| pre-MCP-1d | 40.59 [cA] | 40.68 [bA] | 22.63 [bA] | 31.46 [bB] | 18.88 [bA] | 27.32 [bB] |
| pre-MCP-7d | 39.04 [bcA] | 40.08 [bA] | 20.76 [bA] | 35.29 [cB] | 23.52 [bA] | 26.28 [bA] |
| pre-MCP-10d | 35.96 [bA] | 34.03 [aA] | 29.90 [cA] | 32.14 [bcA] | 23.12 [bA] | 21.69 [aA] |

The means followed by the same lowercase letter in columns and by the same uppercase letters on lines did not differ from one another according to the ANOVA test ($p \leq 0.05$).

As expected, the postharvest 1-MCP application reduced softening in the CTL fruit. Even so, only the fruit from the first and second harvests had values above 20 N after the commercialization period. In the pre-MCP-treated fruit, the postharvest 1-MCP application did not improve the firmness of the fruit harvested on 16 October (Figure 2). Nevertheless, in the fruit harvested later, a higher firmness was shown when the post-MCP treatment was applied.

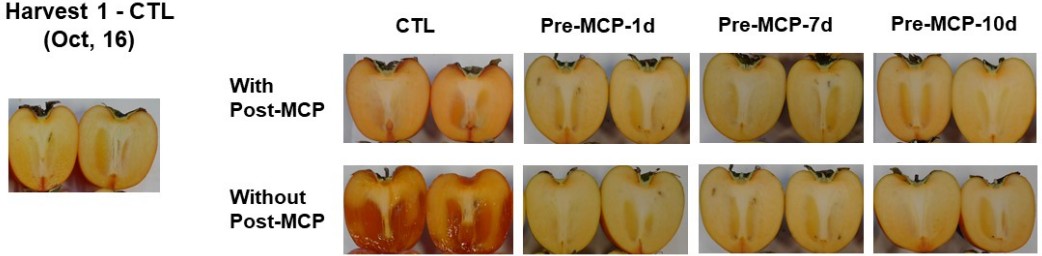

**Figure 2.** Captured images of 'Rojo Brillante' persimmon at harvest (on 16 October) and after the commercialization period (3d at 3 °C plus 6d at 20 °C). Fruit was preharvest treated with 1-MCP applied on days 1 (pre-MCP-1d), 7 (pre-MCP-7d) or 10 (pre-MCP-10d) after ethephon treatment and postharvest treated or not with 1-MCP (post-MCP). CTL is the fruit without preharvest 1-MCP treatment.

### 3.2. Effect of Applying Harvista® after GA on Fruit Quality

The fruit treated once with GA (GA1) without the pre- or postharvest 1-MCP treatments lost firmness throughout storage, with values close to 0 N after 40 days (Figure 3A). As expected, the post-MCP treatment maintained high firmness values, similarly to those of the harvest lasting up to 40 days. After 60 days, a slight decrease in values of 31.5 N was observed. It was noteworthy that the fruit treated with pre-MCP 3 days before harvesting had the same firmness values as the fruit treated with post-MCP throughout the storage period (Figure 4).

Regarding the fruit treated twice with GA (GA2), although those not treated with pre- or post-MCP had higher firmness values than GA1 after 20 days, their values were also close to 0 N after 40 days. The pre-MCP-treated fruit obtained slightly lower firmness values than the post-MCP fruit. After 60 days, both treatments obtained similar values, close to 32 N (Figure 3B).

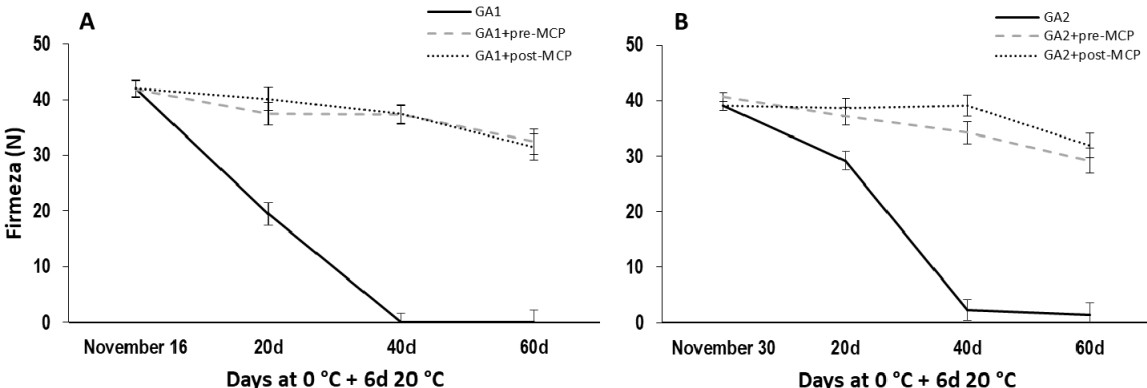

**Figure 3.** Flesh firmness of the 'Rojo Brillante' persimmon treated with 1-MCP during preharvest (pre-MCP), postharvest (post-MCP) or not treated, during cold storage up to 60 d plus a 6-day shelf life at 20 °C. GA1 is the fruit treated once with gibberellic acid (**A**) and GA2 is that treated twice with gibberellic acid (**B**). Vertical bars denote the least significant differences (LSD) intervals ($p \leq 0.05$).

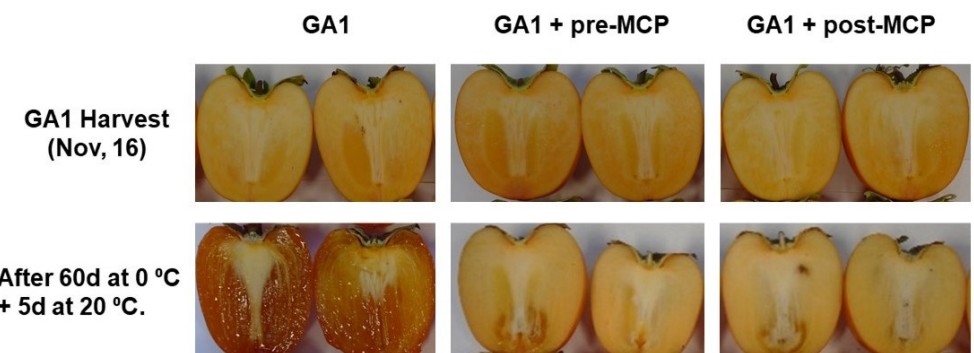

**Figure 4.** Captured images of 'Rojo Brillante' persimmon treated once with gibberellic acid (GA1) at harvest moment (on 16 November) and after 60 d at 0 °C plus 6-day shelf life at 20 °C. Fruit was treated with 1-MCP during preharvest (pre-MCP), postharvest (post-MCP) or not treated.

## 4. Discussion

Most of the studies that evaluate the effect of pre-MCP are made on apples and pears [9,13,14]. The obtained results show that the combination of pre- and postharvest 1-MCP application optimizes the fruit capacity to retain ripening and reduce the incidence of disorders during cold storage, resulting in a higher fruit firmness as well as longer ethylene suppression. In persimmon, only a study on the cultivar 'Fuyu' compared the effects of pre- and post-MCP treatments on the fruit quality during maturity, and positive results were found with the pre-MCP application [10].

In the present study, we found that the pre-MCP application delayed the 'Rojo Brillante' persimmon fruit firmness loss induced by ethephon during the harvest period, and proved to be the most effective treatment when performed 1 day after ethephon application. Therefore, pre-MCP prolonged the harvest period of the ethephon-treated fruit. In papaya, Sañudo et al. [15] found that the application of pre-MCP one day after ethephon was an effective strategy to avoid an excessive softening of the fruit, which allowed for an extension of the shelf-life. It was also reported that the preharvest application of 1-MCP to sweet cherry trees within 3 days of ethephon treatment inhibited ethephon-induced flesh firmness loss [16]. In addition, pre-MCP application has been shown to be a good option for maintaining fruit firmness during the posterior marketing period, when fruit are harvested in mid-October without having to apply a postharvest 1-MCP treatment. Moreover, during the subsequent harvests, the pre- and post-MCP combination maintained a greater flesh firmness during the commercialization period than the single post-MCP application.

On the other hand, gibberellic acid is applied in persimmon fruit to delay ripening and to therefore extend the harvest period [11,17]. 'Rojo Brillante' persimmon destined to cold storage for long periods are those treated on-field with GA and subjected to a post-MCP treatment to avoid the firmness loss. In the present study, a very interesting result is that the application of pre-MCP 3 days before harvesting had the same effect on maintaining the fruit firmness as the post-MCP application during cold storage for up to 60 days.

## 5. Conclusions

The pre-MCP application delayed the 'Rojo Brillante' persimmon fruit firmness loss induced by ethephon, prolonging the fruit harvest period, and proved to be the most effective treatment when performed 1 day after ethephon application. In addition, the pre-MCP application maintained the fruit firmness during the marketing period, when fruit were harvested in mid-October without having to apply a postharvest 1-MCP treatment. Furthermore, during the subsequent harvests, the pre- and post-MCP combination maintained a greater flesh firmness during the commercialization period than the single post-MCP application.

On fruit treated with GA to delay ripening, the application of pre-MCP three days before harvesting maintained the fruit firmness to the same extent as the post-MCP application after cold storage. Thus, replacing the post-MCP application with the pre-MCP treatment can be a very useful tool for improving handling operations in packing houses.

Further studies are necessary to elucidate the role of pre-MCP in maintaining quality during the postharvest persimmon fruit period.

**Author Contributions:** Conceptualization, A.S.; methodology, A.S. and M.V.; formal analysis, N.Q.V. and R.G.; investigation, N.Q.V., R.G., M.V. and A.S.; resources, A.S. and M.V.; data curation, N.Q.V. and R.G.; writing—original draft preparation, N.Q.V.; writing—review and editing, N.Q.V. and A.S.; project administration, A.S.; funding acquisition, A.S. All authors have read and agreed to the published version of the manuscript.

**Funding:** This research was funded by the European Union through the European Regional Development Fund (FEDER) through Project IVIA-51910 and by the company Agrofresh Spain S.L.U.

**Data Availability Statement:** Not applicable.

**Acknowledgments:** N.Q.V. thanks the Spanish Ministry of Science and Innovación for Grant FPI-INIA (PRE2018-085833).

**Conflicts of Interest:** The authors declare no conflict of interest.

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
