# Peer review of "Effect of Preharvest 1-MCP Treatment on the Flesh Firmness of ‘Rojo Brillante’ Persimmon"

_horticulturae, doi:10.3390/horticulturae8050350_

Round 1

Reviewer 1 Report

Line 55, suggests introduction 2.1. Chemicals used in this study;

Line 58, geographical coordinates of the experimental field, maybe also the meteorological conditions!;

Lines 74-91, reformulated, are very difficult to understand;

Line 110, October 20 is the second harvest, line 66 appears October 30 second harvest;

I believe that separate studies should be done on marketing and storage;

I suggest more information from the literature in the discussions

Author Response

-. Line 55, suggests introduction 2.1. Chemicals used in this study;

This information has been added to the revised manuscript (section 2.1).

-. Line 58, geographical coordinates of the experimental field, maybe also the meteorological conditions;

This information has been added to the revised manuscript (lines 58 to 82).

-. Lines 74-91, reformulated, are very difficult to understand;

According to your suggestion, this section has been rewritten for better understanding (section 2.1.2)

 -. Line 110, October 20 is the second harvest, line 66 appears October 30 second harvest;

This has been amended in the revised manuscript (line 120).

-. I believe that separate studies should be done on marketing and storage;

Thank you for your suggestion. It is important to consider that fruit treated with ethephon, at the beginning of the season, is not subjected to cold storage, since the treatment is carried out in order to bring forward the marketing season. In the case of gibberellic acid-treated fruit, the fruit is harvested at the end and middle of the season and the application of 1-MCP is carried out in order to preserve the fruit and extend the marketing season. This is the reason why the postharvest scenarios are different in the two studies.

-. I suggest more information from the literature in the discussions

Further information has been added to the Discussion section (lines 171 to 197) to improve the understanding of the data obtained. New references have been added in the revised manuscript (references 13, 14, 17).

Reviewer 2 Report

Previous studies have revealed that 1-MCP can inhibit ethylene action by binding to ethylene receptors, thus to alleviate chilling injury symptoms in persimmon. This is a given information to solve this issue. The authors expanded preharvest 1-MCP treatment in a famous PVA persimmon cultivar ‘Rojo Brillante’ in EU, and they found that preharvest 1-MCP treatment can maintain the firmness of ‘Rojo Brillante’. The results are interesting, and they presented a strong application of preharvest 1-MCP treatment. As a Communication type article, the results are enough, and they are well-written. I believe that this technique will be widely used to alleviate chilling injury symptoms of persimmon in EU market. Thus, I suggest a minor revision before publication of this manuscript.

Minor comments:

1) If possible, please show ‘Rojo Brillante’ picture before and after preharvest 1-MCP treatment, and also after cold storage. This will give more direct information.

2) If possible, please discuss more about this preharvest 1-MCP treatment. It will be applied in other persimmon cultivation regions.

Author Response

.-  If possible, please show ‘Rojo Brillante’ picture before and after preharvest 1-MCP treatment, and also after cold storage. This will give more direct information.

Two pictures have been added to the manuscript (figures 2 and 4) in case the Editor considers them relevant.

.-  If possible, please discuss more about this preharvest 1-MCP treatment. It will be applied in other persimmon cultivation regions.

Thank you for your comments, further information has been added (lines 171 to 197) to improve the discussion of the data obtained. Nevertheless, there is scarce information on the use of preharvest application of 1-MCP in persimmon.